# Recent Advances on Functional Nucleic-Acid Biosensors

**DOI:** 10.3390/s21217109

**Published:** 2021-10-26

**Authors:** Xinhong Yu, Shiqi Zhang, Wenqiang Guo, Boxi Li, Yang Yang, Bingqing Xie, Ke Li, Li Zhang

**Affiliations:** Department of Animal Nutrition and Feed Science, College of Animal Science and Technology, Huazhong Agricultural University, Wuhan 430070, China; yuxinhong@webmail.hzau.edu.cn (X.Y.); zhangsq@webmail.hzau.edu.cn (S.Z.); gwq@webmail.hzau.edu.cn (W.G.); Rowena78@webmail.hzau.edu.cn (B.L.); yy610143266@webmail.hzau.edu.cn (Y.Y.); xbq@webmail.hzau.edu.cn (B.X.); lico171@outlook.com (K.L.)

**Keywords:** biosensor, biomedicine, functional nucleic acid, mediator

## Abstract

In the past few decades, biosensors have been gradually developed for the rapid detection and monitoring of human diseases. Recently, functional nucleic-acid (FNA) biosensors have attracted the attention of scholars due to a series of advantages such as high stability and strong specificity, as well as the significant progress they have made in terms of biomedical applications. However, there are few reports that systematically and comprehensively summarize its working principles, classification and application. In this review, we primarily introduce functional modes of biosensors that combine functional nucleic acids with different signal output modes. In addition, the mechanisms of action of several media of the FNA biosensor are introduced. Finally, the practical application and existing problems of FNA sensors are discussed, and the future development directions and application prospects of functional nucleic acid sensors are prospected.

## 1. Introduction

According to the latest Global Cancer Statistics, there are about 19.3 million new cancer cases worldwide and nearly 10 million people died of cancer in 2020 [1]. As the largest threat to human life, the early detection of cancer is an effective way to reduce its mortality. In addition, heavy metal poisoning and biological toxins also seriously endanger human health, and their detection methods still have some shortcomings [2,3]. Against this backdrop, biosensors have been developed by integrating modern biotechnology and advanced physical technology. Biosensors are devices that are used for the rapid and sensitive detection of substances at the molecular level. The basic unit of the biosensor includes the identification element, transducer and detector, etc. The components of organisms with molecular recognition capabilities or the organism itself can be used as recognition elements. Traditional DNA biosensors mainly apply nucleic acid molecular hybridization for genetic diagnosis [4]. Due to the emergence of functional nucleic-acid (FNA), the latest FNA biosensors can not only carry out gene detection, but also extend to the detection of metal ions, small molecules, proteins, bacteria, etc. [5,6,7]. It can be credibly stated that their detection range covers all aspects of our life.

In addition to their traditional biological functions such as being carriers of genetic information, functional nucleic acids refer to nucleic acid molecules that have catalytic activity, regulation of gene expression, and specific binding capabilities. They can replace traditional proteases and have independent structures to perform specific biological functions. Up to now, the FNAs used in the biomedical field mainly include aptamers, cleavage ribozymes, mismatch ribozymes, nucleic acid nanomaterials, etc [8,9,10]. With the growth of in-depth research on FNA, researchers have discovered that FNA has many unique functions, such as specific recognition, signal conversion, signal amplification and material assembly [11,12,13,14]. Therefore, FNA biosensors have been widely used in the detection of biomedicine fields. Particularly in terms of the biosensing aspect, scientists have constructed various FNA sensors for different detection targets, which provide a strong guarantee of the continued development of biomedicine.

At present, many domestic and foreign research teams have conducted in-depth and detailed studies on the structure, properties and applications of various FNAs [10]. However, in the past few decades, the following problems still exist in sorting out and summarizing the work in the field of FNA biosensors: there is no inductive overview for the different working principle of FNA biosensors; there is no clear inductive overview of the mediators used in FNA biosensors [15]; and for the application of FNA nanomaterials, there is a lack of detailed introduction and description based on their basic characteristics [16]. Therefore, this review summarizes the working principles and application value of different types of FNA biosensors, and comprehensively explains the significance of mediators in FNA biosensors, as well as the application of FNA biosensors in biomedicine, by investigating the new research that has been conducted in recent years (Table 1). Explaining the unique advantages of FNA biosensors from basic characteristics is vital and valuable for improving the theoretical basis of FNA biosensors and promoting the development and application of FNA biosensors.

## 2. Working Principle for the Detection of Functional Nucleic-Acid Biosensors

There are many types of biosensors, which cannot function without signal transducers. Currently, transmitted signals consist primarily of electrical signals, optical signals, etc (Figure 1). FNA possesses the features of strong specificity, diverse functions and stable structure. Consequently, the detection performance of the FNA biosensors has been clearly improved as compared to conventional sensors.

### 2.1. Electrochemical Functional Nucleic-Acid Biosensors

Electrochemical FNA biosensors are designed by applying FNA to an electrochemical transducer, which converts the analytical information generated by the electrochemical interaction of analytes on the electrode into measurable electrical signals [60]. The electrochemical analysis principle used by this kind of sensor has the characteristics of reusability, high stability, and high affinity. Therefore, it is widely used to detect ions, enzymes, proteins, viruses, and cells [61,62].

An aptamer is a kind of FNA that can be synthesized artificially in vitro. It is screened by the systematic evolution of ligands by exponential enrichment (SELEX) technology, which screens the oligonucleotide fragments that are specifically bound to the target substance from the oligonucleotide library that is synthesized in vitro [63]. Hence, an aptamer has the benefit of customization, and can theoretically be used to detect any target. Aptamers have been widely used as an important recognition element in electrochemical sensors. Using SELEX, Zhang et al screened the H37Rv aptamer of Mycobacterium tuberculosis through SELEX, hybridized it with treated oligonucleotides (AuNPs-DNA) and fixed it on a gold electrode for the detection of H37Rv [17]. When the test sample contains H37Rv, it can specifically bind with the target to release AuNPs-DNA, resulting in a change in the electrode conductivity and the ability to detect H37Rv. Compared with previous studies, FNA sensors can quickly detect H37Rv without culturing or special labeling. Based on a similar principle, electrochemical FNA sensors containing aptamers are also used to detect thrombin, bisphenol A, human papillomavirus, and other substances [64,65,66].

Mucin 1 (MUC1) is an important target that is related to various cancer types, and the development of detection methods has always attracted people’s attention [67,68]. Because surface-enhanced Raman scattering (SERS) has some problems such as poor detection specificity and high equipment requirements [69], a team fixed the MUC1 aptamer on a special electrode that was designed as an electrochemical FNA biosensor for detecting MUC1 [18]. Compared with the SERS method, which is commonly used in hospitals, the sensor offers prominent improvements in the detection range and the detection limit. The detection range of the system is 50–1000 nM, and the limit of detection (LOD) is reduced to 24 nM. It is applied to the detection of human serum samples, and other researchers have further developed many electrochemical FNA sensors for specific tumor cell detection on this basis [19]. Past studies have shown that carbon nanospheres (CNSs) have characteristics such as good chemical stability, bioaffinity, electrical conductivity, and non-biological toxicity [70]. Therefore, they are considered to be ideal materials for electrochemical FNA-sensor reaction platforms. Cao et al. used the MUC1 glycoprotein, which is expressed on the surface of colon cancer DLD-1 cells, as a target to screen specific aptamers and fix them on the CNS reaction platform in order to produce a sensor [19]. Compared with past research, this method greatly reduces the experimental difficulty and has similar detection effects [69]. Furthermore, the CNSs used in the system are non-biotoxic materials, which indicates that this sensor has the potential to be used for real-time detection in the human body. Karpik et al. applied a similar method to design an electrochemical FNA biosensor for detecting prostate cancer cells MUC1 [71]. In the experiment, the sensor detects the MUC1 expression patterns of different prostate cancer cell line models and normal prostate cells, and is able to distinguish between normal prostate cells and different types of prostate cancer cell in clinical testing. Compared with the existing methods such as ELISA and immunoassay, the sensor can resist the interference of a variety of substances, and also has higher detection sensitivity [72,73].

Another major type of FNA is DNAzyme, which has the ability to catalyze nucleic-acid cleavage. A detailed introduction to DNAzyme will be given later. As a result of the extraction of ores and the use of fossil fuels, heavy metal ions, as major environmental pollutants, pose serious threats to human life and health. Recent studies have shown that the electrochemical FNA biosensors developed by DNAzyme have high sensitivity and specificity for heavy metal ions, and thus, they have attracted extensive attention in the field of environmental detection [14,74,75,76]. Tang et al. selected 8–17 DNAzyme for the specific detection of Pb^2+^ [20]. Firstly, the designed substrate chain was hybridized with DNAzyme to form double stranded DNA (dsDNA). Then, DNAzyme was specifically activated by Pb^2+^, which cut and released the substrate chain. Subsequently, the connection probe and signal probe, which can generate electrical signals, were added and cyclically hybridized with DNAzyme to form long DNA structures. Therefore, the long-stranded DNA formed by a DNAzyme was able to contain multiple signal probes, and thus, could significantly amplify the electrical signal. The LOD of Pb^2+^ detected by the electrochemical FNA sensors was reduced to 6.1 pM, which is only one-tenth of that detected by previously examined electrochemical sensors [77,78]. As a result of these improvements in the detection sensitivity and biocompatibility of FNA, the sensor has great application prospects in the detection of human heavy metals.

In biomedical research and food safety investigations, it is often necessary to detect many important small molecular substances, such as adenosine triphosphate (ATP), pesticide residues, etc. However, due to the weight of small molecules, the detection conditions are difficult to ensure with accuracy. By combining aptamers and DNAzyme in aptazymes, one can harness the catalytic cleavage activity of DNAzyme when it has the recognition ability of the aptamer. In addition, aptazymes will only produce catalytic activity when aptamers bind to small molecular targets and cause structural changes [79]. Researchers used the aptamer that specifically recognizes ATP and 10–23 DNAzyme, which are sensitive to Mg^2+^, to form an aptazyme [78]. In addition, the target sequence of DNAzyme was connected with the signal probe to form a catalytic hairpin assembly (CHA). In the experiment, the ATP and Mg^2+^ in the system activated the enzyme activity of aptazyme, and then catalyzed the cleavage of CHA to release DNAzyme and a signal probe. The released Mg^2+^-sensitive DNAzyme continued to participate in the cutting of CHA, resulting in a large number of signal probes being released and bonding to the electrode, which caused an amplification of electrical signals. The electrochemical FNA sensors could accurately detect 0.6 nM ATP in the human serum samples. 

### 2.2. Fluorescent Functional Nucleic-Acid Biosensors

Fluorescent FNA biosensors are composed of a recognition part, a transformation part, and a fluorescent material part. As FNA can be modified by fluorescent groups and quenchants and has strong a recognition ability, fluorescent FNA biosensors are widely used and comprise many types [80]. However, they are mainly divided into labeled fluorescent FNA biosensors and label-free fluorescent FNA biosensors [81]. 

The first report on labeled fluorescent FNA biosensors was published in 2000 [33]. The researchers used a TAMRA fluorophore and a Dabcyl quencher to label the dsDNA, which was composed of a substrate chain hybridized with 8–17 DNAzyme. In a natural state, the quenching effect of the quencher leads to the low-fluorescence reaction of the system. However, in the presence of Pb^2+^, in the above study, 8–17 DNAzyme was activated to cut the substrate chain and then release the substrate chain, which was labeled with fluorescent groups, resulting in the enhancement of the fluorescence signal. Finally, the concentration of Pb^2+^ was obtained by detecting the fluorescence signal intensity, but this method required the labelling of two groups on the enzyme chain, resulting in high experimental cost and time. In recent years, sensors for the detection of metal ions have been further developed. Fluorescein amidite (FAM) was used as a fluorescent group to label the substrate chain, and then hybridized with DNAzyme to form a bulged structure [34]. Then, ethidium bromide (EB) was embedded into the bulged structure as a quencher to quench the fluorescence signal. When DNAzyme was activated by Pb^2+^, the substrate was cut to release EB, so that the fluorescence signal was restored. The sensors made according to this fluorescence turn-on principle did not require the modification of the quencher on the DNAzyme, so the experimental cost was reduced and time was saved. 

In contrast, the FNA sensors designed according to the opposite principle of fluorescence turn-off also showed good performance. Zhang et al used FAM to modify the substrate chain, and then hybridized it with UO_2_^2+^-sensitive DNAzyme to form dsDNA [35]. In the experiment, the fluorescently labeled single-stranded DNA (ssDNA) released by the activated DNAzyme was adsorbed on a special reaction platform, causing the fluorescence signal to be quenched. Compared with the common colorimetric sensor, the FNA sensors designed using this method showed a 200-fold increase in sensitivity to UO_2_^2+^. In addition, the researchers also designed a labeled fluorescent FNA sensor for the detection of H_2_S in the air based on the principle of fluorescence turn-off [37]. Up to now, a variety of metal ions have been detected by labeled fluorescent FNA sensors [82,83,84,85,86,87,88]. 

Label-free fluorescent FNA biosensors are usually designed by using intercalating fluorescent dyes to intercalate FNA, and then changing the fluorescence response through the interaction of the target with FNA, the most common of which is the G-quadruplex structure of FNA [89]. Guanine nucleotides can form a tetrad structure through hydrogen bonds, and multiple such tetrad structures can be stacked to form a G-quadruplex structure [90]. This type of sensor has a similar accuracy to the labeled type, but does not affect the functional activity of FNA. Therefore, it is widely used in many fields. Zhu et al. designed a dsDNA containing a G-rich substrate chain and a DNAzyme that is sensitive to UO_2_^2+^, and used the nucleic-acid-intercalating dye SYBR Green I (SG I) for fluorescent labeling. When DNAzyme was not activated, the fluorescent dye was embedded in dsDNA and showed high fluorescence intensity. When UO_2_^2+^ activated the DNAzyme, the substrate chain was cleaved to form a G-quadruplex, releasing fluorescent dyes, which resulted in a decrease in the fluorescence signal. Compared with the labeled type, the sensors had a similar sensitivity while reducing the difficulty of production [36]. Researchers also applied this principle to the detection of Pb^2+^ environmental samples [91]. As we all know, metal ion pollution in environmental pollution has always been one of the main threats to people’s health and safety. Therefore, a variety of types of label-free fluorescent FNA biosensors have developed that can be used for the detection of K^+^, Na^+^, Ba^2+^, Ir^3+^, Hg^2+^, Tl^+^, Tb^3+^ and other metal ions [92,93,94,95,96,97,98]. The G-quadruplex sequence and the metal cation will form a stable G-quadruplex structure. Screening of the aptamer with the G-quadruplex sequence (oligo-3), which has high selectivity for K^+^, can be applied to K^+^ detection [92]. It is then necessary to select the fluorophore which binds to the generated G-quadruplex. When K^+^ is present, the aptamer is induced to form a G-quadruplex and binds to a fluorophore to generate a high fluorescence reaction. In addition, the G-quadruplex can also be used for Hg^2+^ detection. Hg^2+^ produces a T-Hg^2+^-T mismatch with the DNA sequence [96]. Therefore, the G-quadruplex structure is destroyed, and the fluorescence is turned off. Highly specific aptamers with G-quadruplex sequences and fluorescent groups can be selected for the detection of other metal cations.

In addition, food safety is another major factor affecting human health; therefore, the development and application of label-free fluorescent FNA sensors are also of great value. Taghdisi et al. used aflatoxin B1 (AFB1)-specific aptamers containing a G-quadruplex-forming sequence to form a hairpin structure and used N-methyl mesoporphyrin IX (NMM) as a fluorescent reporter group [11]. It was found that when the aptamer binds to AFB1, it causes the hairpin structure to disintegrate and produce a G-quadruplex structure, and then combine with NMM to produce a strong fluorescent signal. The sensor was able to detect 30–900 pg/mL AFB1 within 30 min. In addition, it was successfully applied in the detection of actual products such as grape juice.

In recent years, in vivo imaging of tumor cells using fluorescent FNA biosensors has gradually become a research hot spot. This technological breakthrough has significantly promoted the development of the biomedical field. Studies have shown that quantum dots (QDs) are a type of nanomaterial with light stability and small size, and are good tools for cell imaging [99]. Chu et al. combined prostate-specific membrane antigen (PSMA) aptamers with luminescent QDs for prostate cancer cellular imaging [38]. The conjugate can target and label living cells, which has important application prospects in the research of human prostate cancer. However, the use of QDs is limited because of their potential biological toxicity and background interference. Therefore, in recent studies, researchers have used carbon dots (CDs) as fluorescent signals to replace QDs. CDs, as a novel member of carbon family, are a type of nanomaterial with good biocompatibility, low biotoxicity and facile preparation. They have become more and more popular in the field of biological detection [100]. CA125, which is an important characteristic antigen of ovarian cancer, was chosen by one group of researchers as the detection target. The team prepared CA125 aptamer-functionalized CDs (CDs-Apta) for cellular imaging of ovarian cancer cells [101]. The experimental results show that the specificity and sensitivity of this method are significantly improved compared with other studies [102,103]. In addition, the cell survival rate was good after 48 h of treatment, indicating very low biotoxicity and the potential for use in patient bioassays. FNA biosensors for tumor cell imaging are being developed towards non-toxicity, high precision and high flexibility. In the future, they will play a key role in tumor mechanism research and drug development.

### 2.3. Colorimetric Functional Nucleic-Acid Biosensors

The colorimetric FNA biosensors are based on the color change reaction of a certain compound, which causes the color of the entire system to change, and the result of the change can be obtained using a spectrophotometer or by visual observation to obtain a quantitative or qualitative result [104]. In order to solve the problem of rapid and high-precision detection of target substances without special instruments, this type of sensor has received extensive attention. Li et al. used aptamers that specifically recognize miRNAs and G-rich DNA (GDNA) to form a hairpin structure [12]. When the aptamer is combined with miRNA, GDNA can combine with hemin to form a G-quadruplex/hemin DNAzyme. This is a common enzyme mimic with horseradish peroxidase (HRP) activity, which can be used to catalyze the conversion of colorless 2,2′-azino-bis (ABTS^2−^) into green ABTS^−^. This design can achieve rapid and highly sensitive detection of miRNA.

Moreover, colorimetric FNA biosensors also show great application potential in the detection of human pathogens. In a recent study, researchers screened out DNAzymes activated by *Helicobacter pylori* (HP) protein and then hybridized the urease-linked substrate strand with the DNAzyme strand [57]. In the experiment, HP protein activated DNAzyme to cleave the substrate chain so that urease was released into the system, and then urease entered the detection area, which contained urea and phenol red, to hydrolyze urea into ammonia, causing pH changes and discoloration of the area. The sensor uses paper as a reaction platform and can be applied to detect HP in human feces, which has important application value in terms of protecting people’s intestinal health. The developed HP detection technologies are mainly divided into invasive and non-invasive tests [105,106,107]. Invasive tests include endoscopic biopsy-based histology, followed by a rapid urease test and molecular PCR. This will consume a lot of time and entail significant costs. Among the non-invasive tests, the urea breath test (UBT) and the stool antigen test are commonly used [108]. However, the result of the non-invasive test is not very reliable and may be affected by urease produced by other bacteria in the digestive tract.

In summary, colorimetric FNA biosensors effectively solve the problem of restricting experimentation due to equipment requirements in the fields of clinical diagnosis and food testing.

### 2.4. Nanotechnology in Functional Nucleic-Acid Biosensors

#### 2.4.1. DNA Hydrogel

Benefiting from the development of self-assembled FNA materials, the same principles can be applied to the construction of DNA hydrogels. DNA hydrogels can be divided into pure DNA hydrogels and composite DNA hydrogels according to whether they contain other non-DNA components [109]. As hydrogels have good biocompatibility and structural variability, DNA hydrogels are widely used in FNA biosensors. Zhang et al. designed a DNA hydrogel formed via the self-assembly of Y-DNA and an aptamer, and embedded AuNPs in it for the detection of thrombin [39]. When thrombin is present, the aptamer binds to it, causing the hydrogel to dissolve. The negatively charged AuNPs and the positively charged fluorescent carrier in the detection system attract each other and produce a quenching effect. The protective effect of the hydrogel makes the FNA biosensors exhibit extremely high stability and specificity in complex samples such as serum. Based on the same principle, the researchers used this hydrogel to develop a colorimetric FNA biosensor for the detection of ochratoxin A in food [110].

In past research, most of the FNA biosensors designed have been disposable products. Improper handling after use will not only cause a huge waste of resources, but also serious harm to the environment. Therefore, in recent research, Mao et al. immobilized the designed soft DNA hydrogel on the surface of the designed reaction platform, and encapsulated HRP to design a set of catalytic systems that can be used for colorimetric analysis [58]. The biggest advantage of this system is that it can be quickly rebuilt after being placed in a buffer solution, and the sensor can be recycled after it is naturally dried. The results of this research have encouraged the development of subsequent recyclable FNA biosensors.

#### 2.4.2. Metal Nanomaterials

Metal nanomaterials have many characteristics such as good biocompatibility, fluorescence performance, activation electrochemical performance, and biocatalytic performance [10]. In recent years, metal nanomaterials have been used more and more widely in the sensors field, which greatly improves the performance of the sensors. With the development of the chemical industry and other industries, Cd^2+^ and polychlorinated biphenyls (PCB), as toxic substances that can induce human cancer and a variety of serious diseases, have posed a major threat to human life and health; therefore, effective detection is of great significance. Recent studies have shown that AuNPs-modified electrodes are used in aptamer electrochemical FNA biosensors for the detection of Cd^2+^ and PCB. Finally, thanks to the advantages of AuNPs such as high dielectric properties and high electron density, the detection sensitivity of the sensors modified by AuNPs has been increased more than 10-fold [21,22]. In addition, silver nanoclusters (AgNCs) also showed great application prospects in environmental monitoring. Studies have shown that when two dark DNA-templated AgNCs are connected by complementary sequences, the fluorescence intensity will increase more than 500-fold [111]. 

Protein kinase is an enzyme that can transfer the phosphoric acid of ATP to polypeptide or protein [112]. The phosphorylation of protein that is regulated by protein kinase is of great significance for the normal physiological function of the protein. Therefore, protein kinase activity (PKA) has become an important physiological and biochemical indicator for serious diseases such as cancer and Alzheimer’s disease [113]. However, its detection process has long been subject to harsh conditions, cumbersome procedures, and high costs. Therefore, Wang et al. connected the ATP-specific aptamer to the dsDNA template (dsDNA-CuNCs) of copper nanoclusters (CuNCs) and used graphene oxide (GO) as the reaction platform [40]. As dsDNA-CuNCs have the characteristics of fluorescence and, due to the π–π stacking effect, GO and ssDNA will spontaneously adsorb and quench the fluorescence reaction, when ATP is combined with the aptamer, the aptamer forms a dsDNA structure so that the system maintains a strong fluorescence reaction. When ATP is converted to ADP, the aptamer is adsorbed to GO because of its ssDNA structure and the fluorescence of CuNCs is quenched. This fluorescent FNA biosensor significantly reduces the experimental difficulty and experimental cost, and the sensitivity meets the requirements of clinical diagnosis.

## 3. Mediators in Functional Nucleic-Acid Biosensors

FNA biosensors are usually composed of FNA, mediators, and vectors. Although mediators are not necessary in FNA biosensors, the use of mediators can help biosensors to detect analytes more sensitively or to optimize the signal generation of biosensors in some specific situations. For example, the mediator can mediate the specific binding process between the sensor and the analyte, mediate the generation of fluorescent or chemical signals, or use its unique properties to significantly increase the sensitivity and detection range of FNA biosensors [41,114]. Currently, there is a wide range of research on FNA biosensor mediators. However, the main focus of the study is on enzyme or metal mediators that are constructed via modification (Figure 2). Generally, enzyme or metal mediators have the advantages of better specificity, higher sensitivity, wider detection ranges, greater ease of construction, and lower prices. The increasing requirements for detection limits in biosensing and medical testing led researchers to pay more attention to the application and research of enzymes or metal mediators in FNA biosensors [27,55,115].

### 3.1. Enzyme-Mediated Mediators

Enzyme-mediated mediators are among the most widespread mediators in FNA biosensors, which play a role in various FNA biosensors. At present, the main enzymes used are deoxyribozyme (DNAzyme), DNase I, exonuclease III (Exo III), terminal deoxynucleotidyl transferase (TdT), S1 nuclease, and APE1. The study of enzyme mediators has promoted the use of FNA biosensors in metal ion detection, biomolecule analysis, food detection, and medicine [41,43,47,59,116,117,118,119].

#### 3.1.1. DNAzyme Based Strategies in Enzyme Mediators

DNAzyme is a kind of synthetic FNA. DNAzyme can easily connect with various fluorescent tags, functional molecules, and solid surfaces, and thus, it is extensively applied in FNA biosensors [42]. DNAzyme is usually constructed as a mediator for FNA biosensors to detect and analyze various substances based on the following two characteristics. One is that DNAzyme has catalytic RNA-cleaving activity. By connecting fluorescent groups and quenchants, DNAzyme-based catalytic beacons can be constructed to generate fluorescent signals. This activity usually requires metal ions as cofactors [120]. The other is that DNAzyme binds to hemin to form G-quadruplexes, mimics peroxidase activity, and connects to affinity ligands for signal transduction [121,122,123]. In recent years, modified DNAzyme has found widespread application as a mediator in FNA biosensors; for example, in environmental monitoring, food safety detection, and liquid analysis [42,115,124].

Yang et al. applied 8–17 DNAzyme to construct a catalytic beacon and used it as a medium for FNA biosensors [41]. Rigorous screening of 8–17 DNAzyme by continuously reducing the concentration of sensitive metal ions in vitro can be used to obtain 8–17DNAzyme with high specificity to target metal ions. Subsequently, the FAM fluorescence group was labeled as the substrate chain of 8–17 DNAzyme, and quenching agents such as DABCYL and BHQ-1S were labeled on the enzyme chain. In the case of the absence of mediator-sensitive metal ions, 8–17 DNAzyme was inactivated, and the hybridization temperature (Tm) between the substrate chain and the enzyme chain was higher than the ambient temperature (T), which ensured the stable hybridization between the enzyme chain and the substrate chain. The fluorescence group could not produce fluorescence signals due to the presence of a quenchant. When sensitive metal ions appeared, the activity of 8–17 DNAzyme was activated, and the substrate chain was cut precisely into two fragments. As the Tm values of the two fragments fell below the substrate chain, dehybridization between the enzyme chain and the substrate chain resulted in the release of fluorescent groups and the separation of quenching agents. Simultaneously, a fluorescence signal was generated and used for the detection of metal ions. In addition, by adding a second quenchant to the 8–17 DNAzyme substrate chain, the signal-to-noise ratio increased more than 10-fold. FNA biosensors composed of this kind of mediator solve the problems of the narrow range and low sensitivity of the traditional sensor in metal ion detection [42].

In another work, Xue et al. designed an enzyme-mediated material based on oxime chemistry-assisted DNAzyme for the detection of Lipopolysaccharide (LPS) with FNA biosensors [43]. The 3′ end of the ESI of the DNAzyme chain was modified with the aminoxy group, and the 5′ end of the ESII was modified with the aldehyde group to make the two chains react under mild conditions to form a complete enzyme chain with an oxime bond. Then, the fluorescence group and the fluorescence quenchant were connected at the 3′ and 5′ ends, respectively, of the substrate chain and connected with the enzyme chain to form a hybrid chain. Mg^2+^ was used to activate the enzyme chain and cut the product chain to release the fluorescent group. In the presence of LPS and galactose oxidase (GAox), LPS was specifically modified by GAox to produce a 6’- terminal aldehyde group and then reacted with the 3′-terminal ammonia oxygen on ESI, which prevented the formation of the chain and led to the inability to produce fluorescence. This change in fluorescence intensity allowed for sensitive detection and analysis of LPS. Meanwhile, when using cresol red, phenol red, and neutral red as indicators, it was shown that the detection limits of the colorimetric method were within acceptable limits, and this superior versatility was the most striking feature of the method [43].

Direct use of the catalytic activity of DNAzyme can also construct mediators and apply them to FNA biosensors. For example, a method of target-induced activation was designed by using two hairpin structures. One of the hairpin structures contained DNAzyme active fragments and had the function of recognizing BLM, while the other could be catalyzed and cleaved by DNAzyme. In the presence of BLM, the Fe^2+^ in the environment could be used to form a complex and cut the hairpin structure to release DNAzyme. Then, Zn^2+^ was added to activate DNAzyme; the activated DNAzyme was catalytically cleaved another fragment released from the hairpin structure to enable the adsorption of the metal–organic framework (MOF), and it formed a FNA biosensor by connecting MOF with an electrode that could sensitively detect BLM. Compared with the use of DNAzyme to construct the mediator formed by the catalytic beacon, this method exhibited enhanced selectivity, better convenience, and excellent specificity in the application of FNA biosensors. In particular, this method showed an impressive detection limit of 4 pM for BLM [23]. 

Liu et al. used DNAzyme to simulate catalase activity to construct mediators [13]. They found that building g-quadruplex/heme DNase as a mediator had a similar effect to using catalase as a mediator. Afterwards, they used G-quadruplex/hemin DNAzyme catalytic hydrogen peroxide (H_2_O_2_)-oxidized ABTS^2−^ to enhance the absorption intensity of the sensor at 415 nm. Signal amplification was accomplished in this way to improve the detection sensitivity. The most satisfying advantage of the FNA biosensor constructed using this mediator over conventional ELISA assays was its high detection limit of 0.01 U/mL for the assay of T4PNKP. 

#### 3.1.2. DNase I-Based Strategies in Enzyme Mediators

In addition to using DNAzyme to construct mediators, deoxyribonuclease I (DNase I) has also been applied as a mediator in FNA biosensors. DNase I is an enzyme that can hydrolyze ssDNA and dsDNA [117]. DNase I has often been avoided in conventional sensors due to the properties that hydrolyze the nucleic-acid chain. As the FNA biosensor research field has developed, so too has the use of DNase I as a mediator. Researchers have found that a stronger signal strength can be obtained by using DNase I to degrade FNAs. With the degradation of FNAs, the analyte is released and then combined with probes and other recognition substances. This process helps in obtaining repeated signals to reach the signal amplification. This signal amplification strategy is a cost-effective approach to highly sensitive detection [32,44,125,126,127]. 

To optimize the fluorescence signal of the AuNCs-aptamer FNA biosensor, Guo et al. designed a fluorescence amplification strategy based on the DNase I mediator. Using the DNase I activity of the mediator to catalyze the degradation of aptamers that had produced fluorescence signals and to release the patulin (PAT) for detection, the released PAT continued to bind to aptamers and produce fluorescence signals. The signal amplification was fulfilled by repeating this process [44]. 

In a recent study, a DNase I mediator was developed for the amplification of fluorescence signals. The study confirmed that this mediator improved the signal intensity of a magnetic graphene oxide-assisted homogeneous electrochemiluminescence aptasensor. Based on the catalytic activity of DNase I, the mediator could assist with the cyclic dissociation and degradation of the free aptamer to implement signal amplification. The results showed that the selectivity and specificity of FNA biosensor for target analytes were significantly improved compared with classical methods, and the detection limit reached 4 pg/mL [32]. 

#### 3.1.3. Exo III-Based Strategies in Enzyme Mediators

When using DNase I enzyme and DNAzyme to construct a FNA biosensor, the cofactors are usually used to help construct the mediator due to the sequence dependence of DNase I enzyme and DNAzyme. The disadvantage is most obvious in the signal amplification strategy, which also limits the application of these two enzymes as mediators [44,116,128,129]. Exonuclease III (Exo III) is a sequence-independent enzyme, so it is unnecessary to consider specific enzyme recognition sites in application [118]. Exo III was deployed early on for DNA sequencing because of its substrate preference for dsDNA, with fewer than four single nucleotides at the flat end or the 3′ convex end [130]. With the emergence of FNA biosensors and the continuous improvement of Exo III research, the signal amplification strategy for FNA biosensors based on Exo III-assisted cyclic amplification has become a research hotspot. The use of Exo III as a mediator for auxiliary signal amplification can be applied to all kinds of FNA biosensors to perform accurate and rapid detection of target substances [24,45,131]. 

For instance, Wu et al. employed Exo III to construct a mediator to actuate the signal amplification of the functional nuclear sensor [45]. Hairpin DNA specifically recognized the target DNA. Exo III released the DNA by digestion, then another hairpin structure and Exo III were used to mediate cyclic amplification in order to produce many walking strands (WS). In Exo III-mediated WS, AgNCs-conjugated magnetic beads (AgNCs-MBS) liberate significant numbers of AgNCs for fluorescence detection in order to acquire the signal amplification of the cascade amplification. The problem of poor signal when using AgNCs as fluorescence signal is solved by Exo III, which greatly improves the detection sensitivity of the FNA biosensor without fluorescence group labeling. Furthermore, this FNA biosensor has been experimentally demonstrated to detect the cancer-related gene p53. 

In addition, using the driving target recovery characteristics of Exo III also constitutes a method for using Exo III in FNA biosensors. Zhang et al. used Exo III as a mediator to release the target material Hg^2+^ from the DNA probe [24]. Then, the free DNA probe bound the Hg^2+^ again and amplified the signal through the catalytic precipitation reaction of DNAzyme. This step was repeated to obtain a stronger current to detect the Hg^2+^. The FNA biosensor constructed by this method possessed an ultra-high sensitivity to Hg^2+^, with a detection limit of 0.1 nM. Furthermore, the method showed great potential for practical applications.

#### 3.1.4. The Application of TdT in Enzyme Mediators

Terminal deoxynucleotidyl transferase (TdT) is a DNA polymerase that catalyzes the binding of deoxynucleotides to the 3′ hydroxyl terminal of DNA or RNA molecules [132]. Similar to Exo III, TdT is a sequence-independent enzyme with no special requirements for substrate sequences. dsDNA or ssDNA with a 3′ protruding end, 3′ concave end, or flat end can be used as primers to initiate the extension reaction catalyzed by TdT, and the highest catalytic efficiency is that of the 3′ protruding end. Adjusting the content of dNTP in the reaction environment can determine the sequence of extension products [133].Since the extension mechanism of the TdT enzyme usually needs to have the 3′ protruding end because of its characteristics, the current research direction is towards the generation of the exposed 3′-OH as the substrate of TdT enzyme extension by restriction enzyme digestion, or the design of a particular sequence, thus expanding the scope of application of the TdT enzyme as a mediator in FNA biosensors [46,119,134]. 

Xu et al. designed a method for sensitive miRNA detection, using the TdT enzyme as a mediator to assist with target recovery [46]. In the presence of miRNA, the 3′-phosphorylated DNA probe (DNA-3′-PO_3_) targeted miRNA to form DNA/RNA double strands and activated double-stranded nuclease (DSN) to specifically cleave the DNA in the double strands. After cleavage, the target miRNA was released and combined with another DNA-3′-PO_3_ to amplify the signal by cyclic reaction, and then a DNA fragment with 3′-OH was obtained by enriching the product. On the other hand, the TdT enzyme used enriched DNA fragments as primers and extended them using dTTP to obtain a product with an ultra-long poly T-tail that combined with Cu^2+^ to form poly-TCuNPs. This process enabled the fluorescence detection of miRNA using FNA biosensors. The FNA biosensor constructed using this mediator enabled the genotyping of TP53 SNPs in a linear range of 50–1000 nM and with a detection limit of 0.11 nM. 

In addition to using the target recovery of TdT enzymes to construct mediators in FNA biosensors, TdT enzymes can also be used in signal amplification strategies. Another group proposed a dual-signal amplification strategy to construct TdT enzyme mediators for the detection of thrombin, raising the detection limit to 0.31 pm. The thrombin was bound to the probe with DNAzyme activity to form a complex. Then, the complex combined with the oligonucleotides of DNAzyme was substrated on the electrode and cleaved by DNAzyme to produce exposed 3′-OH. The TdT enzyme was activated by 3′-OH and mediated the elongation reaction. At this time, a significant number of specific G-quadruplex products was obtained by controlling the content of dATP and dGTP in the environment to accomplish the first step of a signal amplification. G-quadruplex and hemin combined to form G-quadruplex/hemin to further amplify the signal [119]. 

#### 3.1.5. S1 Nuclease Based Strategies in Enzyme Mediators

S1 nuclease is a highly single-strand-specific endonuclease derived from *Aspergillus oryzae*. It was first isolated from *Aspergillus oryzae* by Vogt in 1973 [135]. The primary function of S1 nuclease is to degrade ssDNA or RNA, but its activity is not vigorous when double-stranded substances such as dsDNA and double-stranded RNA are used as substrates [135,136,137]. Therefore, the S1 nuclease is often used as a mediator in FNA biosensors to take advantage of the unique single strand degradation characteristics [59,138,139].

To detect the single-nucleotide polymorphisms (SNPs) of the TP53 gene, Xu et al. designed a method involving the use of S1 nuclease and peptide nucleic acid (PNA) as mediators to assist hemin-functionalized single-walled carbon nanotubes (hemin-SWCNTs) [59]. The FNA biosensor constructed using this mediator enabled genotyping of TP53 SNPs in a linear range of 50–1000 nM and with a detection limit of 0.11 nM. In their work, the sequence of PNA was designed to match the DNA to form PNA/DNA double strands and avoid the degradation of the S1 nuclease. In the meantime, the S1 nuclease digested and degraded the base mismatch DNA. The color change of ssPNA or PNA/DNA double strands was produced by hemin-SWCNTs with peroxidase-like activity in the TMB/H_2_O_2_ system. This color signal was utilized for the colorimetric analysis of FNA biosensors. Additionally, this kind of mediator can monitor all kinds of nucleic acids by changing the PNA sequence. Compared with traditional mediators, it has the advantages of simple operation, high sensitivity, and accurate results.

#### 3.1.6. APE1-Based Strategies in Enzyme Mediators

Apurinc/apyrimidinic endonuclease 1 (APE1) is one of the essential enzymes in the APE family. It has both apurinc/apyrimidinic sites (AP site) endonuclease activities and the ability to regulate the DNA binding activity of transcription factors. APE1 plays a vital role in the base repair pathway of DNA damage. It can cut the AP site produced by glycosylase, digest the DNA chain from the 5′ end, and then repair DNA damage with the DNA polymerase and DNA ligase [140,141]. According to the cleavage activity of APE1 and its involvement in DNA damage repair, it is usually used in FNA biosensors to detect the activity of DNA glycosylase [25,47,48,49]. 

In their research, Wang and coworkers have been committed to using APE1 as a mediator in FNA biosensors for detecting the activity of DNA glycosylase [47,48,49]. In 2016, they developed a method to explore the enzyme activity of DNA glycosylase human 8-oxoguanine-DNA glycosylase 1 (hOGG1), using the cleavage activity of APE1 on the AP site as a mediator. They used biotin to modify the sense chain of substrate DNA, while the antisense chain was altered by damaged guanine 8oxoG. As a result of adding hOGG1 to cut and release 8oxoG, meanwhile, an AP site was generated to initiate the base excision repair pathway. Then, the APE1 specifically recognized the AP site and exerted catalytic activity to produce a mononucleotide chain gap. With the aid of deoxyribonucleic acid polymerase β, the Cy5-labeled, dGTP was catalyzed to bind to the substrate gap to acquire DNA with Cy5 marker modification. This DNA could bind to a single quantum dot (QD) surface via biotin. Concurrently, fluorescence resonance energy transfer (FRET) from OD to Cy5 generated a signal to analyze the activity of hOGG1 by total internal reflection fluorescence (TIRF) imaging [47]. Then, in 2018 they constructed a more sensitive strategy for detecting hOGG1, in which used two-molecular-beacon-mediated autocatalytic self-replication-driven cascaded recycling amplification and a hairpin substrate with damaged 8oxoG. Activating the APE1 by added hOGG1, they cleaved the AP site to form prominent 9-nt DNA sequences. Subsequently, autocatalytic self-replication-driven cascaded recycling amplification was activated under the co-mediation of FOK I, and the 9-nt DNA sequences produced a large number of fluorescence signals to detect hOGG1 activity [47,48]. More recently, a method designed for the real-time monitoring of DNA glycosylase activity was presented by the same group. A controllable catalytic cracking-mediated fluorescence recovery strategy based on T7exo was constructed by using APE1 as a mediator. In the presence of human alkyladenine DNA glycosylase (hAAG), hairpin probe 1 (HP1) recognized hAAG, and the hAAG exerted glycosylase activity at the damaged 2-deoxyinosine site of the hairpin probe to obtain the AP site. The AP site was cleaved by APE1 and formed DNA double strands with DNA polymerase and DNA ligase. Then, the hybridization of DNA double strands and hairpin probe 2 (HP2) induced T7exo-assisted autocatalytic recycling signal amplification to receive a strong fluorescence signal for the detection of hAAG. The FNA biosensor constructed by this strategy can be applied to monitor DNA glycosylase hAAG activity in real time [49]. 

Zhang et al. also used APE1 as a mediator in FNA biosensors [25]. They designed an APE1-mediated FNA biosensor that uses a-cyclodextrin-functionalized gold/silica cell-shell nanoparticles to detect the activity of DNA glycosylase hOGG1. The addition of a-cyclodextrin-functionalized gold/silica cell-shell nanoparticles added more recognition sites and dramatically enhanced the recognition sensitivity. The electrochemiluminescence FNA biosensor constructed by this method had a detection range of 2–200 UmL^−1^ for hOGG1. 

Additionally, APE1 was considered to be a biomarker for cancer detection, and it was demonstrated that APE1 was often characteristically overexpressed in a variety of cancer cells such as hepatocellular carcinoma cells and lung cancer cells [142,143,144,145]. This property makes APE1 suitable as a medium for constructing medically relevant FNA biosensors for cancer detection, targeted delivery of cancer drugs, and cancer cell screening [50,146]. In 2016, Zhai et al. reported an even more novel example of cancer cell detection. They designed silica-coated magnetic nanoparticles (SiMNP) based on the physiological properties of APE1. Upon detection, APE1 selectively cleaved the AP site of the avidin-oriented abasic-site-containing DNA strands (AP-DNA) on the SiMNP surface, which resulted in the separation of the fluorescent moiety from the quencher on AP-DNA-SiMNP. The detection of APE1 expression could be performed by measuring the intensity of the released fluorescent signal. They successfully constructed an FNA biosensor to detect APE1 expression in different living cells through AP-DNA-SiMNP, thus achieving rapid and sensitive differentiation between ordinary cells and cancer cells [50]. The study also proposed that linking cancer-treatment-related drugs with AP-DNA-SIMNP could greatly optimize the targeted delivery of drugs. The reason was that the high level of APE1 in cancer cells allowed AP-DNA-SiMNP to rapidly target drug delivery to specific cancer sites [142]. In this way, a more scalable and implementable optimization solution for cancer drug delivery was proposed. In a similar vein, Zhang and coworkers proposed a method to construct an FNA biosensor for cancer detection using APE1 as a mediator through nanocomposites [146]. The nanocomposite material was formed by the combination of single-molecule DNA and graphene quantum dots, which could be used as a probe to bind APE1 through the AP site [50]. After the combination, significant accumulation of fluorescence signals in living cells was generated through repeated enzyme-catalyzed cycles. Ultimately, cancer screening could be performed by measuring the difference in signal intensity [146]. Compared to biosensors constructed by means of the other methods published thus far, the FNA biosensor constructed using this method was able to screen the same type of cells under different cellular conditions [147]. 

### 3.2. Metal-Mediated Mediators

In the use of traditional sensors, metals are usually used as substances to be detected. However, with the development of research in this field, metal-type mediators have gradually entered researchers’ field of vision. At present, the main kinds of metal type mediators are metal–organic frameworks (MOFs), metal nanomaterials, and ionic metals [148,149,150,151,152]. Using metal-type mediators in FNA biosensors can provide high sensitivity, high flexibility, and a wide detection range. Therefore, they have been widely used in the fields of environmental monitoring, food safety, molecular biology, and medicines [27,55,114,153]. 

#### 3.2.1. The Application of MOFs in Metal Mediators

The MOFs formed through metal–ligand coordination were first proposed by Yaghi in 1995. By designing the interaction between metal nodes and organic ligands, the structure of the MOF can be changed to achieve different functions. Thus, MOFs are commonly used in the fields of molecular adsorption and gas separation [148,154,155]. With the advent of FNA biosensors, the application MOFs as mediators to assist sensors has become a research hotspot. Several studies have revealed that MOFs have been applied in signal amplification strategies, auxiliary probe labeling, and fluorescence quenching [26,51,52].

In the utilization of FNA biosensors, MOFs can facilitate FNAs to achieve signal amplification. Xiangdan et al. constructed a method using MOF as a mediator that enables highly sensitive miRNA imaging. They formulated an ATP aptamer loaded with cancer cell membrane (CM)-encapsulated glutathione (GSH)-responsive metal−organic framework (MOF) nanoparticles (NPs) (MOF-H1/H2@CM). The ATP aptamer prevented bio-orthogonal activation of the heterogeneous chain reaction (HCR) in the absence of endogenous ATP. After the miRNA was added, GSH mediated MOF-H1/H2@CM degradation to release the HCR element. Then, endogenous ATP was expressed to bind to the aptamer, which activated the HCR functional element to drive signal amplification. Finally, high-sensitivity imaging of miRNA was achieved [51]. 

In addition, MOF can also perform the function of fluorescence quencher as a mediator in FNA biosensors. The high catalytic activity of fluorophore-labeled Y-shaped DNAzyme/3D MOF-MoS_2_NBs allowed them to be used for the high-sensitivity detection of Hg^2+^, Ni^2+^, and Ag^+^. In the presence of the metal ion to be detected, the DNAzyme activity of the Y-shaped DNAzyme was activated to cleave the substrate strand. Then, the quenching of fluorescent groups by 3D MOF-MoS_2_NBs resulted in a change in the fluorescence signal. Ultimately, fluorescence analysis enabled the sensitive and accurate detection of metal ions. This FNA biosensor achieved excellent sensitivity compared to previously published sensors of the same type and reached detection limits of 0.11 nM, 7.8 μM, and 0.25 nM for Hg^2+^, Ni^2+^, and Ag^+^ [52]. 

In a recent study, an Fe-MOFs mediator was developed for the detection of thrombin concentrations with an impressive limit of detection of 59.6 fM. A nucleic-acid aptamer signal probe (SP) label with Fe-MOFs was developed in this study, and Fe-MOFs signals were then generated by the recognition of thrombin. Meanwhile, the signal-stabilized [Fe(CN)_6_]^3−/4−^ electrolyte solution was used as the inner reference probe (IR). Eventually, the ratio of Fe-MOFS-SP and [Fe(CN)_6_]^3−/4−^-IR signals could be used to indicate the prothrombin concentration [26]. 

In another work, an Ag- and Zn-based bimetallic MOF (Ag/Zn-MOF) construction method was proposed. Ag/Zn-MOF could be co-mediated with cyclic voltammetry (CV) and electrochemical impedance spectroscopy (EIS) for electrochemical signal amplification. This method was demonstrated to assist with the capture probe of HCV in the construction of a HCV FNA biosensor with a detection limit of 0.64 fM [27]. 

A novel FNA biosensor based on a Cu-porphyrin (Cu-TCPP)/G-quadruplex-hemin nanocomposite was proposed by Ma et al. Cu-TCPP with peroxidase activity was synthesized using the surfactant-assisted method coupled with G-quadruplex-hemin DNAzyme for the purpose of catalyzing the reduction of H_2_O_2_. Furthermore, these two substances could be used to perform “biological barcode” signal amplification. At present, studies have shown that the concentration imbalance of H_2_O_2_ could be used as a biomarker in the current clinical medicine to detect myocardial infarction as well as atherosclerosis [156]. This method offered a new application idea for the detection of H_2_O_2_ in clinical medicine. Moreover, it solved the problems of high cost, long consumption time, and ease of interference in the traditional detection methods [157,158].

#### 3.2.2. The Application of Metal Nanomaterials in Metal Mediators

Metal nanomaterials have certain advantages as mediators in some fields due to their excellent optical, electrical, and catalytic properties. Metal nanomaterials have been widely used in various sensors, which play an essential role in the fields of environmental detection, food safety, and nanomedicine [149,150,151]. Many of the recent studies on FNA biosensors have been focused on the use of metal nanomaterials as mediators. Metal nanomaterial mediators can enhance the accuracy and selectivity of the sensors. In addition, the morphology of metal nanomaterials is variable according to the needs of the application, thus allowing for an extended range of detection types [29,53,159]. 

For instance, a complex (A_S-T7_) formed by silver nanoclusters (AgNCs), a scaffold sequence (A_S_), and a T-rich sequence (A_T7_) can efficiently detect organic mercury (CH_3_Hg^+^ and C_2_H_5_Hg^+^). In the absence of organic mercury, the reduced Ag^+^ combined with A_S_ to form A_S_-templated AgNCs. This conjunction ensured that the fluorescence signal from AgNCs was effectively generated. When organic mercury CH_3_Hg^+^ and C_2_H_5_Hg^+^ were present, the organic mercury bound to A_T7_ and then quenched the fluorescence by means of photoinduced electron transfer (PET). Eventually, organic mercury could be effectively monitored by detecting the change in the fluorescence signal [53].

In work performed by Lee’s group, aptamer-modified gold nanorods (GNRs) were introduced as mediators in FNA biosensors to optimize localized surface plasmon resonance (LSPR) in in situ sample analysis in order to pinpoint inaccuracies. They constructed an optical-fiber-based LSPR aptasensor for the detection of ochratoxin A (OTA). The GNRs bound OTA upon specific recognition of OTA by the aptamer. Meanwhile, the aptamer transformed from a single strand to a G-quadruplex (GQx) structure. The generation of GQx changed the local refractive index (RI) of GNRs, which resulted in a spectral red shift of the LSPR peak. This shift could be used to detect the OTA. A detection limit as low as 12.0 pM (3 S) was achieved for OTA in this work. Compared with the OTA detection method commonly used in the medical field, the FNA biosensor established by this paper possessed a more convenient usage and basically satisfied the limits of medical detection. Meanwhile, this paper presented a strategy for the rapid completion of in situ testing, which provided a more convenient testing concept for the current clinical medical testing of OTA [159]. 

Additionally, metallic nanomaterials such as Au nanoparticles (AuNs) were shown to be able to be used as mediators to construct highly sensitive signal probes for FNA biosensors. Functionalized porphyrin-based covalent-linked nanomaterial OAPs-Por (OAPs-Por/Thi@AuNPs-ssDNA) constructed by conjugating Thi, Au nanoparticles, and single-stranded DNA was used as a probe for the precise and responsive detection of uracil-DNA glycosylase (UDG). Subsequently, the probe was modified by hairpin DNA (hDNA) with four uracils. In the presence of UDG, it mediated the unfolding of the hDNA structure and generated single-stranded DNA hybridization with the probe. At the same time, Thi generated an electrochemical signal, which was amplified by OAPs-Por and used to detect UDG [29]. 

Wang et al. introduced an anti-EpCAM (AE) and triggered DNA-modified gold nanostar (AuN)-assisted aptamers via the recognition of circulating tumor cells (CTCs) to build a dual recognition mechanism for the detection of CTCs. This mediator, with a dual recognition mechanism, could be cooperatively used with surface-enhanced Raman scattering (SERS) to build CTC FNA biosensors. The limit of SERS detection reached 5 cells/mLw with the use of this method, and the limit of fluorescence detection was 10 cells/mL [54]. 

#### 3.2.3. The Application of Metal Ions in Metal Mediators

With the continuous development of metal mediators, metal ions have become a hot research topic. In recent years, there has been an increasing amount of literature on metal ion mediators. Researchers have found that the exertion of the T–Hg^2+^–T mismatch reaction of Hg^+^ can stabilize two complementary single-stranded DNAs, and thus, can be applied to enhance the signal. Moreover, the binding properties of the metal ion can be used to construct fluorescent signal-generating structures or recognition probes. Currently, metal ion mediators are frequently used in environmental monitoring, food quality testing, and biomedical applications [55,56,152,160].

The toxicity of excessive melamine can cause serious harm to children’s health. However, the current detection methods used in clinical medicine are still unable to detect melamine with high specificity [152]. Hg^2+^ possesses excellent melamine binding ability. Thus, this metal ion can specifically detect melamine. Wang et al. proposed a method to generate a fluorescence signal by using Hg^2+^ as a mediator to create a T–Hg^2+^–T mismatch reaction-assisted N-methylmesoporphyrin IX (NMM)/G-quadruplex [55]. Then, they designed a FNA biosensor for the detection of melamine based on this method. They constructed two oligonucleotide DNA strands (P1 and P2), in which P1 possesses a self-assembled structure to form a G-quadruplex. P1 and P2 formed a double strand through a Hg^2+^-mediated T–Hg^2+^–T mismatching reaction and G–C base pairing. The double chain structure prevented the self-assembly of the G-quadruplex, ensuring that no signals were generated in the absence of the detected substance. When melamine was present in the sample, the melamine was first mixed with the Hg^2+^ solution to form the Hg^2+^–melamine complex, which resulted in a significant decrease in the Hg^2+^ concentration. Low concentrations of Hg^2+^ failed to mediate the T–Hg^2+^–T mismatch reaction. Thus, P1 and P2 were unable to form a double-stranded structure, which led to the self-assembly of G-quadruplex. Finally, NMM was incorporated to bind the G-quadruplex and generated a fluorescent signal for the detection of melamine content. 

Ma and coworkers proposed a method to construct a rapid, sensitive, and highly specific label-free fluorescence FNA biosensor and demonstrated experimentally that this method could detect miRNAs with high sensitivity [56]. The sensor constructed by this simple method reached a detection limit of 25 fM and showed a linear response between 50 and 500 fM. They built Terpyridine (L) and terpyridine-Zn(II) (ZnCl_2_L) complexes using Zn^2+^ as a mediator. When miRNA was present, branched rolling circle amplification (BRCA) was initiated to produce massive amounts of pyrophosphate (PPi). Then, ZnCl_2_L was bound to PPi to produce fluorescence intensity, which indicated the concentration of the target miRNA. 

In other work, Cu^2+^ was used as a mediator in the construction of a pyrophosphatase (PPase) FNA biosensor. A G-quadruplex-Cu^2+^ DNAzyme construction method was designed based on the fact that Cu^2+^ can coordinate with pyrophosphate (PPi) to form a Cu^2+^–PPi compound. The presence of PPase specifically degraded the PPi in the Cu^2+^–PPi compound to produce free Cu^2+^. Then, Cu^2+^, coupled with G-rich DNA, it mediated the formation of a G-quadruplex-Cu^2+^ DNAzyme with peroxidase activity. Finally, the catalytic reduction of H_2_O_2_ by peroxidase activity generated a current signal for detecting PPase activity [30]. 

The highly sensitive FNA biosensor mediator, using a combined method of capturing and probing peptide nucleic acid (PNA) and coordinating the interaction of Zr^4+^, was introduced by Wang et al. They constructed a functional microelectrode using PNA and Zr^4+^ [31]. It was demonstrated that the FNA biosensor based on this microelectrode could be used in nucleic-acid detection and clinical analysis. 

## 4. Functional Nucleic-Acid Biosensors Design for Biomedicine

### 4.1. Functional Nucleic-Acid Biosensors Design for Real-Time Imaging in Living Cells

In recent years, with the continuous development of the biomedical field, non-invasive live detection has gradually become a hot topic. The understanding of FNA biosensors has continuously deepened, and its application potential in biological detection has been continuously explored. To date, many teams have made great progress in the application of FNA biosensors for biological detection (Figure 3).

The first study was focused on the living cell imaging of neurotransmitter release at the single living cell level. Zeng et al. experimentally verified the proposed concept with dopamine (DA) as the target [161]. The researchers designed an aptamer-modified amphiphilic DNA-nanoprism structure. The top three vertices of the structure have poly-A ssDNA extension, and the corresponding bottom three vertices extend cholesterol labels. Due to the hydrophobic interaction between cholesterol and the phospholipid bilayer of the cell membrane, a DNA-nanoprism was inserted and fixed on the surface of nerve cells within 10 min. The DA aptamer was modified with FAM and contained a poly-A ssDNA hybridization sequence hybridized with ssDNA, which was modified with black hole quenching group 1 (BHQ1) to form dsDNA; it quenched the FAM fluorescence reaction. The prepared dsDNA was bound to the DNA-nanoprism. When DA was released, the aptamer combined with DA to release the quenching group BHQ1, making FAM recover the fluorescence signal. Therefore, the release of DA in a single living cell can be successfully detected by fluorescence signal imaging, which proves the feasibility of the strategy and its potential application in the study of neurological diseases [161].

Xiong et al. applied genetically encoded fluorescent protein (FP) to the intracellular imaging of Mg^2+^ and Zn^2+^ [162]. When implementing their methods, principle is to select the 10–23 RNA cleavage DNAzyme that specifically recognizes Mg^2+^ or the 8–17 RNA cleavage DNAzyme that specifically recognizes Zn^2+^ to cut Clover2 mRNA and inhibit the expression of green FP (GFP). The Ruby2 gene mRNA corresponding to the corresponding red FP (RFP) is not affected. Therefore, DNAzyme and plasmids expressing Clover2 and Ruby2 are co-transfected into HeLa cells, and finally the distribution of target ions in the cells can be observed by confocal microscope imaging or flow cytometry.

### 4.2. Functional Nucleic-Acid Biosensors Design for Pathogen Detection

Disease detection is currently the most researched and most widely used field of FNA biosensors in biomedicine. FNA biosensors are mainly used to detect the level of pathogenic bacteria and virus content.

The effective diagnosis of pathogenic bacteria is of great significance in the process of disease prevention, control, and diagnosis, in order to meet the needs of rapid diagnosis of methicillin-resistant *Staphylococcus aureus* (MRSA). A research group coupled the MRSA marker protein penicillin binding protein (PBP2a) aptamer to graphene oxide-loaded gold nanoparticles (GO/Au) [163]. Preliminary experiments proved that GO/Au has H_2_O_2_ enzymatic activity, which can catalyze the conversion of TMB to make the solution blue and significantly increase the catalytic activity after binding to ssDNA. Therefore, when the aptamer is not bound to the target, the spontaneous combination of the aptamer and GO/Au increases the H_2_O_2_ enzyme activity and makes the solution blue due to the π–π stacking effect. The color of the system changes according to the concentration of the target [163]. A sensor prepared in this way can detect at least 20 nM of PBP2a.

In addition, high-efficiency biosensors are also needed to assist with diagnosis in virus detection. Based on the principle of the aggregation of AuNPs induced by NaCl, an efficient fluorescent sensor for detecting hepatitis B e antigen (HBeAg) was designed. The aptamer of HBeAg was screened out and added to the AuNPs solution containing NaCl. The combination of the aptamer and AuNPs with NaCl caused the inhibited solution to change color from purple to red. After adding HBeAg-positive serum, the aptamer was bound to HBeAg to release AuNPs, and the aggregation solution turned from purple to red. The colorimetric FNA biosensor could complete the detection of HBeAg in serum samples within 2 min and the LOD met the clinical diagnostic criteria for hepatitis B [164].

### 4.3. Functional Nucleic-Acid Biosensor Design for Body Fluid Index Detection

In modern medicine, the diagnosis of diseases is inseparable from the detection of physiological and biochemical indicators of body fluids. According to different purposes, it is usually necessary to detect different complex samples in addition to serum, such as urine and semen. Therefore, in order to meet the needs of different samples, Chen et al. designed a Cu^2+^-sensitive DNAzyme to form a fluorescent FNA biosensor for the determination of histidine content in urine [165]. First, they used BHQ1 to modify the 5′ end of the DNAzyme and the substrate chain, and then used FAM to modify the 3′ end of the substrate chain. After the DNAzyme chain was hybridized with the substrate chain, the fluorescence of the whole system was quenched. When Cu^2+^ was present, the enzyme chain was activated to cleave the substrate and release FAM to produce a green fluorescent signal. When histidine and Cu^2+^ existed at the same time, the two combined to form a complex, and thus, the DNAzyme was not be quenched by the fluorescence in the activated system. For the analysis of experimental results, the FNA biosensor has good clinical value, but also the disadvantage of needing to use a fluorescent group label [165].

In their research on the use of FNA biosensors for the detection of semen, Sun et al. applied a zirconium metal–organic framework (Zr-MOF) to improve the colorimetric FNA biosensor array in terms of semen detection [166]. By coupling ssDNA-modified AuNPs to Zr-MOF, Zr-MOF would adsorb ssDNA-modified AuNPs and cause them to precipitate. In this array, six ssDNA sequence-modified AuNPs were selected to distinguish 10 proteins in semen. After adding semen to the system, the effect of different components in the semen on the ssDNA-modified AuNPs was also different, which in turn affected the precipitation of AuNPs and made the solution show different colors. Based on the color change of the solution, the sample could be diagnosed quickly and accurately [166]. In the future, this method can be applied to disease diagnosis by changing the ssDNA sequence.

### 4.4. Functional Nucleic-Acid Biosensors Design for the Detection of Cancer Targets 

In the clinical diagnosis of prostate cancer (PCa), vascular endothelial growth factor (VEGF) and prostate-specific antigen (PSA) are used as target substances for detection using electrochemical sensors that are improved by aptamers and nanomaterials. Graphene oxide/ssDNA (GO-ssDNA) assembled by VEGF aptamer and graphene oxide is wrapped on the electrode. During detection, the aptamer will bind to VEGF and then add nanoparticles designed with VEGF and PSA antibodies, and then the antibodies will specifically bind to VEGF so that the PSA antibody can capture the PSA in the sample after the nanoparticles are immobilized on the electrode. Based on the principle that the target binding on the electrode leads to conductivity that causes the electrical signal to change, two kinds of PCa markers can be efficiently detected. The detection time of this method is only 1h, and the detection range is similar to that of the traditional ELISA method. The LOD of VEGF is 50 pg/mL, and the LOD of PSA is 1ng/L [167]. This is also the first electrochemical FNA biosensor that can detect two cancer markers in a short time with the help of aptamers.

## 5. Concluding Discussion and Perspectives

Nowadays, there are many kinds of FNA sensors, which are widely used in biomedicine fields. Compared with traditional DNA sensors, the detection time and sensitivity are greatly improved, allowing them to play a significant role in the detection of various human diseases. Their future development direction focuses on portability (for example, paper-based sensors) and reusability (for example, hydrogel sensors) [168,169]. In addition, most of the current FNA sensors can only be used to detect a single target, and only a few of these can be used to detect multiple targets at the same time. It is possible that a portable sensor for the simultaneous and efficient detection of multiple targets can be developed in combination with chip technology and programming technology [170].

Additionally, the use of mediators in FNA sensors has not been popularized enough; most of them are still in the laboratory stage. It has been reported in many studies that the use of mediators in FNA sensors can play a role in biomedicine, the environment, and other fields, but there are few examples of the application of mediators in actual detection, either in literature reviews or in studies of specific construction methods [16,109,171]. Although the earliest electrochemical sensors for detecting blood glucose have been developed for three generations and widely used in blood glucose detection in vitro and in vivo, most FNA sensors cannot be put into practical applications. The properties of some enzymes and metal ions will also affect their practical applications in FNA biosensors. Some metal ions are toxic and some enzymes require suitable active factors and temperature, which restrict their input into practical applications [172,173]. Therefore, the current mediators used in FNA sensors for vitro detection are more common. However, it is more difficult to construct mediators and they need more operations for in vivo detection.

The application of real-time imaging technology to biosensors is a fairly cutting-edge research area in the field of biomedicine. On account of the capacity of the technology to observe the specific trends of the disease in a small time span, it allows researchers to have a general direction in treatment and a more accurate basis for disease development prediction. For instance, the real-time imaging biosensor can be used to observe the condition of the patient’s diseased area at all times during a surgical operation, which can determine the success of the operation [174]. In the clinical research of live animals, real-time imaging can always pay attention to various changes in the animal body, which greatly shortens the time of clinical research [161]. Many technologies support real-time imaging research nowadays. For example, computers with advanced computing power can solve complex problems involving imaging data, whereas, with the gradual development of the current medical research towards the fields of cells and molecules, the detection limits of many biosensors used for real-time imaging can no longer meet the needs of researchers. The technology for high-precision and high-sensitivity imaging of small molecules is not yet perfect. For example, real-time mRNA imaging in cells is focused on dynamic cells, but resolution is inversely proportional to the speed of cell movement, and increasing the detection sensitivity of biosensors can solve this problem [175]. Therefore, the construction of high-sensitivity biosensors is one of the most pressing problems at present. The addition of the mediator can greatly increase the sensitivity of the FNA biosensor, thereby obtaining a higher-definition imaging effect. At the same time, it can also greatly reduce the imaging time and the probability of imaging errors. Therefore, the combination of FNA biosensors and real-time imaging technology for disease detection will be a fairly promising direction in biomedicine.

## Figures and Tables

**Figure 1 sensors-21-07109-f001:**
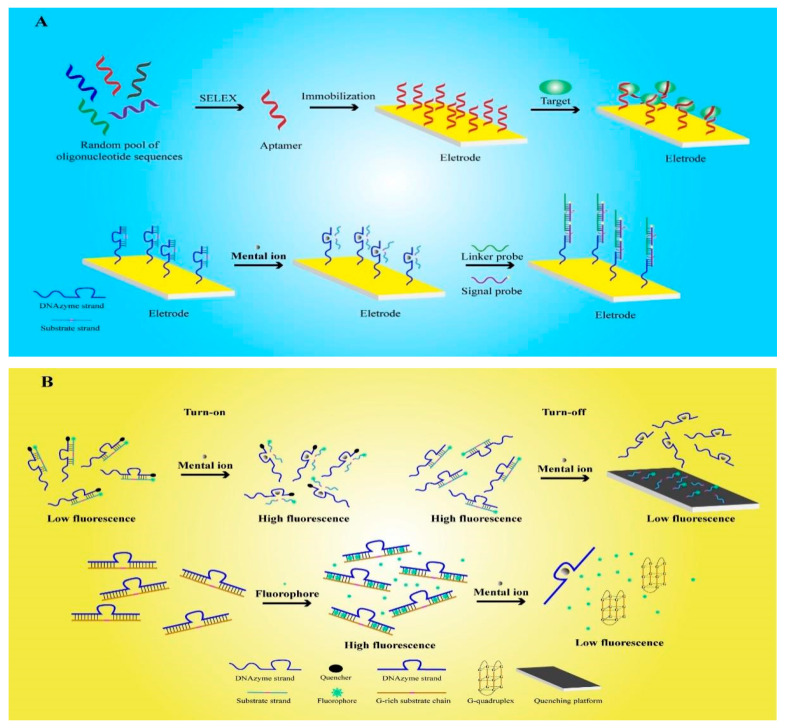
Different working principle of functional nucleic-acid (FNA) biosensors. (**A**) Electrochemical FNA biosensor: The aptamer is screened in terms of the systematic evolution of ligands by exponential enrichment (SELEX) and fixed on the electrode to bind to the target. DNAzyme is immobilized on the electrode and then specifically recognizes the metal ion which activates it. After cutting the substrate chain, the signal is amplified by connecting the linker probe and the signal probe. (**B**) Fluorescent FNA biosensor: The turn-on principle and turn-off principle of the labeled fluorescent FNA biosensor designed by DNAzyme. The working principle of the label-free fluorescent FNA biosensor designed by DNAzyme and G-quadruplex. (**C**) Colorimetric FNA biosensor: The working principle of HRP mimics composed of G-quadruplex and hemin. The working principle of the colorimetric FNA biosensor composed of DNAzyme and urease.

**Figure 2 sensors-21-07109-f002:**
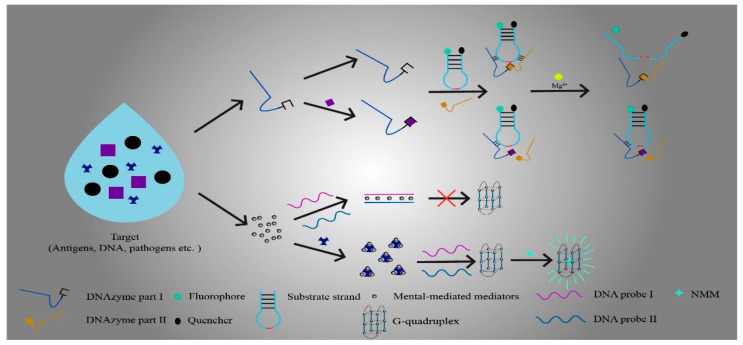
The enzyme-mediated mediator causes structural changes by binding to the target, which corresponds to the different reaction processes of the FNA biosensor. The metal-mediated mediator binds to the target and causes the structure of the DNA probe to change in response to different detection results.

**Figure 3 sensors-21-07109-f003:**
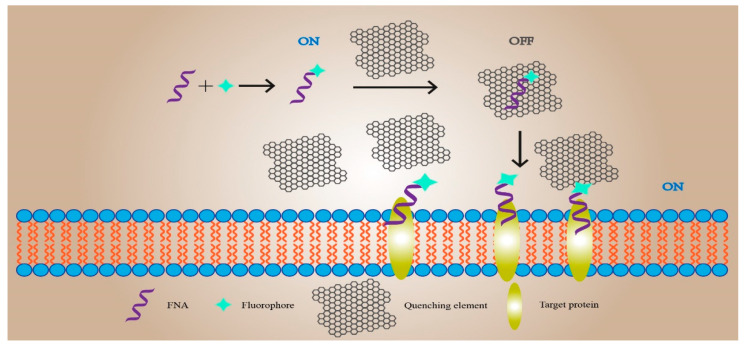
The FNA-modified fluorophore combines with the quenching element and is transported into the organism. By encountering the target protein, FNA binds to the target protein and detaches from the quenching element because of its different affinity for the performance of real-time imaging in living cells.

**Table 1 sensors-21-07109-t001:** Overview of the practical applications of different functional nucleic-acid (FNA) biosensors.

Classification	Mediators	Test Substances	Applications	Reference
Electrochemical FNA biosensor	AuNPs-DNA	H37Rv	Early Clinical Diagnosis of Tuberculosis	[17]
	MUC1 aptamer	MUC1	Cancer Cell Screening in Clinical Testing	[18,19]
	DNAzyme	Pb^2+^	Detection of Pb^2+^ in River Water Samples and Clinical Testing	[20]
	Aptazymes	ATP	Detection of ATP in Human Serum Samples	[14]
	AuNPs	Cd^2+^ and PCB	Water Sample Analysis in Environmental Testing	[21,22]
	DNAzyme	BLM	Pharmaceutical analysis and clinical sample	[23]
	Exo III	Hg^2+^	Water Sample Analysis in Environmental Testing	[24]
	APE1	hOGG1	Detection of hOGG1 Activity in Clinical Diagnosis	[25]
	Fe-MOFs	TB	Quantification of Thrombin in Clinical Medicine Detection	[26]
	Ag/Zn-MOF	HCV	Detection of Real Human Samples in HCV Diagnosis	[27]
	Cu-TCPP	H_2_O_2_	Chemical industry and Clinical Medicine	[28]
	AuNs	UDG	Clinical Diagnostics and Biomedical Research	[29]
	Cu^2+^	PPase activity	Quantification of PPase Activity in Clinical Analysis	[30]
	Zr^4+^	Nucleic acids	Nucleic Acids Detection and Clinical Analysis	[31]
	DNase I	PAT	Highly Sensitive Detection of Marine Toxins	[32]
Fluorescent FNA biosensor	8–17 DNAzyme	Pb^2+^	Detection of Pb^2+^ in Environmental Pollution Monitoring	[33]
	DNAzyme	Pb^2+^	Detection of Noxious Ions in Aqueous Solutions	[34]
	DNAzyme	UO_2_^2+^	Detection of UO_2_^2+^ in Aqueous Environment	[35,36]
	DNAzyme	H_2_S	Clinical Toxicology Monitoring	[37]
	Aptamer	AFB1	Evaluation of Grape Juice and Human Serum Samples Spiked with AFB1	[11]
	PSMA aptamers	Tumor cells	In Vivo Imaging of Tumor Cells	[38]
	AuNPs	TB	Thrombin Assay in Complex Protein Samples	[39]
	dsDNA-CuNCs	PKA	PKA Assay for Cell Lysates in Clinical Medicine	[40]
	DNAzyme	Pb^2+^	Toxic Ion Detection	[41,42]
	DNAzyme	LPS	Quantification of LPS in Food	[43]
	DNase I	PAT	Clinical Toxicology Monitoring	[44]
	Exo III	DNA	Clinical Diagnosis	[45]
	TdT	miRNA	Clinical Diagnosis	[46]
	APE1	hOGG1	Biomedical Research and Clinical Diagnosis	[47]
	APE1	hOGG1	Biomedical Research and Clinical Diagnosis	[48]
	APE1	HAAG	DNA repair-related biochemical research, Clinical Diagnosis, Drug Discovery, and Cancer Therapy	[49]
	APE1	Cancer cells	Cancer Detection, Targeted Delivery of Cancer Drugs and Cancer Cell Screening	[50]
	CM-encapsulated GSH-responsive MOF NPs	miRNA	Spatiotemporal and Bioorthogonal Imaging of miRNAs In Vitro and In Vivo with High Sensitivity.	[51]
	3D MOF-MoS_2_NBs	Hg^2+^, Ni^2+^, Ag^+^	Analysis of Real Water Samples with Interfering Contaminants	[52]
	AgNCs	Organic mercury	On-site Detection of Organic Mercury in Seafood and Other Biological Samples	[53]
	AuNSs	CTCs	CTCs Detection Platform for Clinical Diagnostics	[54]
	Hg^2+^	Melamine	Detection of Melamine in Raw Milk and Milk Powder	[55]
	Zn^2+^	miRNA	Biomedical and Clinical Diagnosis	[56]
	G-quadruplex/hemin DNAzyme	microRNA	Clinical Molecular Diagnosis of microRNA	[12]
Colorimetric FNA biosensor	DNAzyme	HP	Human Pathogens Monitoring, Clinical Diagnosis	[57]
	G-quadruplex/hemin DNAzyme	Hydrogen peroxide and bilirubin	Detection of Hydrogen Peroxide and Bilirubin in Serum	[58]
	G-quadruplex/hemin DNAzyme	T4PNKP	Drug Discovery	[13]
	S1 nuclease	DNA	Clinical Diagnosis, Medical Science	[59]

## Data Availability

Not applicable.

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
