# Peer review of "Recent Advances on Functional Nucleic-Acid Biosensors"

_sensors, 2021, doi:10.3390/s21217109_

Round 1

Reviewer 1 Report

The manuscript described the functional nucleic-acid (FNA) biosensors for detection of biomedicine. In this overview, the authors first introduced functional modes of biosensors which combine functional nucleic acids with different signal output modes. They also summarized the action mechanism of several media of FNA biosensor and their application in biomedicine. Additionally, they discussed the practical application and existing problems of FNA sensors, and the future development direction and application prospects of functional nucleic acid sensors. To publish in sensors, the authors need to have a broad scope. I recommend publication after addressing the following concerns.

1.There is only one picture in the manuscript, some other representative pictures need to be added and discussed in the manuscript.

2.The “Introduction” mainly emphasized on the status of FNA sensors and and the application of FNA biosensors in biomedicine in the form of a table comparison. Some brief description of such applications will be helpful.

3. Some discussion on aptamer-based electrochemical FNA biosensing for tumor cell detection needs to be added in the “2.1. Electrochemical functional nucleic-acid biosensors”.

4. Some discussion on fluorescent functional nucleic-acid biosensors for imaging analysis and detection of tumor cells needs to be added in the “Fluorescent functional nucleic-acid biosensors”.

5. The authors are encouraged to review more examples related to real biologically important samples and problems associated with it.

6. I recommend the authors to read the manuscript once again and correct many misleading phrases and mistakes.

Reviewer 2 Report

Referees’ Comments

  • No.: sensors-1398091
  • Title: Recent advances on functional nucleic-acid biosensors for detection of biomedicine
  • Journal: Sensors

The manuscript entitled “Recent advances on functional nucleic-acid biosensors for detection of biomedicine” by Yu et al. reviewed the state-of-art functional nucleic-acid biosensors for biomedical applications. Overall the manuscript is well prepared and very interesting. Therefore, I recommend its publication in Sensors after minor revision. Following are the issues that need to be addressed:

Specific comments:

1. It is better to add more information with additional references about the 3.1.6 APE1 based strategies in the enzyme mediators section. A few references have only used this section.

2. This is the reviewer’s concern. Even though this manuscript is a review article, there are a few figures throughout the manuscript. This review includes exciting topics in many sections. However, it is difficult to understand the explanations because they are too wordy and lack of corresponding figures. Therefore, the authors are encouraged to give new 2-3 figures with related explanations.

3. The authors should reorganize Table 1 based on the working principles. Because the working principles were mixed in Table 1, it is not easy to understand intuitively.

Reviewer 3 Report

General comment:

The article summarized recent advances of functional nucleic acid (FNA)-based biosensors. It briefly introduced the generic design of FNA-based biosensors and thoroughly reviewed the individual working mechanism of different types of biosensors. This work could provide comprehensive insight into the field. 

Revision suggestions:

  1. Please consolidate the structure of the paper and focus on one topic without discursiveness. 
    This manuscript is entitled "Recent advances on functional nucleic-acid biosensors for detection of biomedicine", which indicates a topic focusing on the development and utility of the biosensor on biomedicine discovery. However, the first paragraph of the introduction section specifically introduced the urgent need for cancer early screening and the significance of biosensors on cancer diagnosis. Nonetheless, the whole paper thoroughly described the development and working mechanism of sensors of various irrelevant areas, such as pollution monitoring, food industry, and chemical engineering. The whole statement is not well-organized and confuses readers. 

  2. Please further discuss the performance and impact of the sensors covered by this paper.
    The paper comprehensively covered the technical details of each biosensor, which is impressive. However, since the article proposed to discuss how the development of biosensors facilitating biomedicine discovery, the performance and impact of a developed biosensor on biomedicine discovery should be highlighted as well. Why is this biosensor novel and innovative? How could it help the detection of biomedicine? How much does it outperform the state of the art? These questions should be addressed with corresponding details (e.g. accuracy improvement, better sensitivity/specificity, precision/recall, etc.).

Reviewer 4 Report

The review presents useful revision of new analytical approaches based on the use of functional nucleic-acids. The paper accords to thematic field of the Sensors journal and may be published in it after some revisions:

  1. The manuscript contains a lot of examples of analytical developments for food control, environmental control etc. The authors' comments that these tasks affecting human health (line 184) are very broad, and really these examples are not subjects of biomedicine. Moreover, practically all considered approaches have universal character and may be applied to detect various target analytes independent on their action on human health. So, the title may be simply reduced to «Recent advances on functional nucleic-acid biosensors».

  1. The first paragraph of the review is overloaded by trivial Wikipedia-style statements («people desire to live longer and enhance quality of life», « DNA biosensors use deoxyribonucleic acid molecules» etc.) which are redundant in the article for a professional audience. This paragraph can be significantly shortened without loss for the subsequent presentation of the material.

  1. Strong definition of functional nucleic-acids should be given at the beginning of the article. Comments at lines 38-39 present some features of FNA, but do not fix border of this term.

  1. As well as «recent advances» are indicated in the title, the authors should clarify time interval or some other rules for the choice of the presented examples finalizing their introduction (lines 53-59).

  1. Lines 161-162 «Compared with the traditional method, the FNA sensors designed using 161 this method showed a 200-fold increase in sensitivity» The considered traditional method is not clear from this text.

  1. Considering FNA biosensors for metal cations (page 7); it would be useful to comment their specificity. What features of interaction allow sensors to detect one strictly defined cation?

  1. Lines 205-213. Urease tests for Helicobacter pylori are known in medical diagnostics, and their common formats are realized without DNAzymes. What are the benefits of using DNAzymes for this specific task?

  1. Lines 245-247 «Among them, silver nanoclusters (AgNCs), copper nanoclusters (CuNCs), and AuNPs are the most widely used metal nanomaterials.» The criteria for this assessment are unclear. For AuNPs, it coincides with the overall statistics of existing developments, while the choice of the other two metal nanomaterials is debatable. It is recommended to exclude this categorical assessment from the article.

  1. Line 258 «Protein kinase is a kind of biological enzyme». It will be better to reduce this statement to simple «Protein kinase is an enzyme».

  1. Line 275 «FNA biosensors are usually composed of FNA, mediators, and vectors» The necessity of vectors is not clear from the previous text. Probably, only a part of FNA biosensors need in vectors. The use of mediators in optical biosensors is also non-typical demand. In any case, the term «mediator» has different senses in various scientific fields and so should be defined in the review.

  1. Line 307 «Yang et al. applied 8–17dnazyme». 8–17DNAzyme should be typed here.

  1. Lines 326-327. «Xue et al. designed a more economical, convenient, and intuitive … material ». Please don't repeat authors' self-assessments without sufficient evidence of these estimates.

  1. Line 335 «LPS was specifically catalyzed». Maybe, modified?

  1. Lines 580-581. «Metal nanomaterials have excellent optical, electrical, and catalytic properties compared to traditional materials». A strange statement. A lot of applications still use Metal nanomaterials have excellent optical, electrical, and catalytic properties com-580 pared to traditional materials without their change to metal nanomaterials.

  1. At lines 596-598 the authors correctly consider modification of aptamer (special oligonucleotide) by a nanoparticle. However, at lines 588-590 a total complex of silver nanoclusters (AgNCs), a scaffold 588 sequence (AS), and a T-rich sequence (AT7) is curiously named as aptamer.

  1. The review was written practically without figures. It complicates understanding of the describer processes and needs revision along all the manuscript.

Round 2

Reviewer 4 Report

The manuscript has been successfully revised and now may be accepted.